# Why Integrated Reporting? Insights from Early Adoption in an Emerging Economy

José Navarrete-Oyarce [1,*], Hugo Moraga-Flores [2], Juan Alejandro Gallegos Mardones [3] and José Luis Gallizo [4]

1   Facultad de Economía y Negocios, Universidad Andrés Bello, Concepción 4030000, Chile
2   Departamento de Contabilidad y Auditoría, Universidad de Concepción, Concepción 4030000, Chile; hmoraga@udec.cl
3   Facultad de Ciencias Económicas y Administrativas, Departamento de Auditoría y Sistemas de Información, Universidad Católica de la Santísima Concepción, Concepción 4030000, Chile; jgallegos@ucsc.cl
4   Departamento de Administración de Empresas, Universidad de Lleida, 25002 Lleida, Spain; joseluis.gallizo@udl.cat
*   Correspondence: jose.navarrete@unab.cl

**Abstract:** The purpose of this research is to contribute new and relevant information about the early adoption of integrated reporting in an emerging economy, in this case the Chilean economy, with emphasis on the reasons for its use, the perceived benefits, and the difficulties experienced during the process of adoption. Methodologically, this work uses a qualitative approach, utilising semi-structured interviews as a data collection instrument that are applied to the managers in charge of preparing this reporting for a sample of companies that trade on the Chilean stock exchange. These interviews were interpreted using an analysis of their content. The results obtained are in accordance with the literature and the empirical evidence, with the characteristics belonging to an emerging economy and highlighting the importance of communicating with the stakeholders. The relationship with the stakeholders and the inclusion in international contexts were the most valued. It is also highlighted that all of the companies analysed declare sustainability as a strategic focus, which is an infrequent situation in developing economies. One limitation of this work is the low valuation and use that market agents still assign to integrated reporting in Chile, which affects the degree of preparation and utilisation, and could be explained by other types of incentives and motivations.

**Keywords:** integrated reporting; emerging economy; sustainability; revelations; company social responsibility

## 1. Introduction

Current tendencies leave business secrecy behind, with transparency as the prevailing paradigm. In this way, stakeholders demand more and more strategic information, with presence in financial and non-financial aspects [1,2]. This phenomenon has diverse explanations from a theoretical point of view. On one hand, legitimacy theory suggests that companies communicate information to their interest groups, with the aim of meeting social expectations [3]. In addition, stakeholder theory is based on the company's obligation to relate to diverse actors that surround it and could have interests in it [4]. Finally, agency theory explains the voluntary revelation of information through the concept of asymmetries of information [5]. However, in this scenario of abundance of information and multiple reports, it becomes difficult to establish a global image for the company [6,7].

As a consequence of the financial scandals of the early 2000s and recent financial crises, the elaboration and use of new models of reporting corporate information have been encouraged, considering strategic, social, economic and environmental aspects [8]. In accordance with this, integrated reporting (IR) aims to inform the potential of the company to create value in the short and long term through a strategic vision of the entity [9,10], and

has, starting from the year 2011, been transformed into an innovative information tool for interest groups [11].

The IR collects and incorporates the current limitations of traditional financial reports that tend to be criticised for their length and lack of articulation [12]. For this reason, the IR strengthens the non-financial information, increasing the transparency of the organisations and their capacity to create value [13–16]. From the founding of the International Integrated Reporting Council, IIRC, which goes back to December 2009, the Prince of Wales, in his speech asking for the creation of this organism, made it clear that the main objective of the IIRC was to "Create a globally accepted framework for the sustainability accounting, for helping to develop more complete and understandable information on the total performance of an organisation, both prospectively and retrospectively, to meet the needs of a new, more sustainable global economic model" [17].

From its beginnings, the IR has been voluntary and many times with each organisation's own focus due to the absence of a general norm [18,19]. Because of this, in 2011 a pilot programme was started that sought to create an adequate normative framework [15–20], and that would later begin to massify the IR at an international level [19]. This increase in the adoption of a report that incorporates material related to environmental, social, and corporate governance, ESG, aspects has gone hand in hand with the obligatory requirement for some companies in the European Union in relation to divulging non-financial and diverse information, exempting those companies that emit a report with the information requested [21]. South Africa has gone even further, as it required the obligatory preparation of the IR for certain companies [22]. Chile has not remained exempt from this tendency and through NCG N°461, emitted by the Commission for the Financial Market, CMF, obliges the entities under its supervision to report the policies, practices and goals adopted in environmental, social and governance or ESG material [23]. This is born from the growing relevance that the divulging of information in relation to these policies, practices and goals has acquired at a local and international level [16]. In the same way, it requests the inclusion of indicators of sustainability, of regulatory compliance, and of the salary gap, etc. Thus, the new Annual Statement is structured based on an integrated reporting focus.

The objective of this work is to analyse the early adoption of the IR in an emerging economy, in this case the Chilean economy, with emphasis on the reasons for this adoption, the perceived benefits, and the difficulties experienced during the process. This research makes significant contributions to the current literature, as it is focused on an emerging market where sustainability has not reached high levels of compliance with respect to international norms, fundamentally because of the lack of development of sustainability strategies from the companies and the scarce influence of public regulatory bodies [24,25].

The originality of this work is given by the fact that the study was carried out in an emerging economy, such as the Chilean economy, located in Latin America, a region that is not so present in the global discussion of integrated reporting. [26] In this regard, it is important to note that most of the literature has studied the relationship between the behaviour of companies and their environmental performance in what we call Western economies, although in recent years the study has also been extended to other geographical areas, such as emerging countries.

Currently, Latin America demonstrates a growing trend in improving the transparency and use of the information issued by companies due to the effects on performance [27], and for this reason, it seems relevant to us to study the value of the IR into Chilean companies, as this country has demonstrated sustainable economic development since the mid-1990s [28,29]. For this reason, this paper presents a contribution to the global discussion on integrated reporting and its application.

This article is structured in five parts. The introduction is the first part, then the review of the literature. The methodology and sample, then the results and discussion, and finally, the conclusions, follow.

## 2. Literature Review

Ecological disasters, social imbalances, financial crises and financial scandals have provided evidence of the fragility and lack of information provided by companies to allow an adequate evaluation of their sustainability in surroundings that are more and more complex every day [10,30–32]. This has led people to question the quality of the information given to interest groups and has motivated companies to voluntarily prepare sustainability reports as a way of complementing financial information and showing greater social and environmental commitment [8,10,33,34].

The aforementioned has not been free of problems, as it is often difficult to establish relationships between the financial and non-financial information prepared by the companies when it is revealed independently and without articulation [6]. It is in this context that the preparation and use of the IR has been encouraged, as it considers strategic, social, economic and environmental aspects that affect business decision making [7–9]. It is for this reason that the IR recognises the limitations of the information and the lack of interaction between financial and non-financial information [12], favouring the organisation's transparency and their capacity to create value [13,14,35], as well as an adequate evaluation of the true drivers of corporate value [36]. Table 1 shows the evolution of the presentation of reports emitted by companies in recent decades, beginning from an exclusively financial focus towards a report of the sustainability of the company.

**Table 1.** Evolution of integrated reporting about financial and non-financial information.

| Chronology | Report Types | Dimensions Reported |
|---|---|---|
| Before the 1970s | Financial reports. | Financial dimension exclusively. |
| Decades of the 1970s and 1980s | Financial reports isolated from social and/or environmental information. | Financial dimension exclusively. Considers only one dimension of sustainability (social and/or environmental). |
| Decade of the 1990s | Financial reports. Sustainability reports considering social and environmental information. | Financial dimension exclusively. Considers two dimensions of sustainability (social and environmental). |
| Decade of the 2000s | First annual reports that seek to integrate financial and sustainability information. | Considers a valuation of the financial and non-financial information in only one context. |
| Current trend | Integrated reporting. | Considers a valuation of three dimensions of sustainability, financial, environmental and social, in only one context. |

Source: Adapted from [37].

As indicated, the IR offers strategic information to interest groups, allowing a better evaluation of the results of an organisation in the long term [16,38]. However, some organisations see this type of initiative as a greater cost instead of as an obligation to better report to their stakeholders [39]. Given the growing value assigned to the IR by financial analysts, due to the positive effects on organisations' performance, its importance has increased significantly in recent years [40–43]. [44] finds similar results, observing a premium in the valuation of those companies that use the IR as a means of communicating information with their interest groups, and observing an increase in share price due to the better quality of accounting information in the IR [45].

New demands in the preparation and presentation of the IR imply a change of attitude and relationship of the companies with their interest groups [10,37,42,46], and must not be limited to a change in form, but also in purpose, which should be to positively affect the internal processes and organisational structure of the companies that drive value creation [9,10]. Therefore, the IR incorporates a flexible focus in its preparation that permits an adequate evaluation and comparison of companies [9–47]. For more detail about this, Table 2 is incorporated.

**Table 2.** Guidelines for preparing integrated reporting statement.

| Guiding Principles | Content Elements | Fundamental Concepts |
| --- | --- | --- |
| 1. Strategic focus and future orientation; 2. Connectivity of information; 3. Stakeholder relationships; 4. Materiality; 5. Conciseness; 6. Reliability and completeness; 7. Consistency and comparability. | 1. Organisational overview and external environment; 2. Governance; 3. Business model; 4. Risks and opportunities; 5. Strategy, resource allocation and performance; 6. Outlook; 7. Basis of preparation and presentation. | Value creation for the organisation and for other stakeholders. |

Source: Adapted from [9].

There exists evidence that shows that the process of decision making improves through the use of the IR [48], as well as showing a positive and significant relationship between the IR and company performance [49,50], as it complements the companies' financial information with a greater organisational complexity with multiple business segments or with important investments in intangible assets [51–53].

There also exists evidence that demonstrates that the use of the IR reduces agency problems between the insiders and suppliers of external capital as a product of asymmetries of information, which would negatively affect the realisation of new investment projects. For this reason, it is observed that companies that provide information to interest groups through the IR show a greater valuation, as it will allow them to make more precise cash flow estimations [12,48,53,54]. [53] suggests that the divulgation of information related to the environmental performance, governance, interest group participation, social and ethical considerations and identification of risks and opportunities derived from the business model and strategy can be transformed into a source of competitiveness. In addition, there exist other benefits from the use of IR, which can be appreciated in Table 3.

**Table 3.** Benefits of the adoption of the integrated reporting.

| Dimensions | Benefits |
| --- | --- |
| Internal | Improves comprehension of performance indicators. Clarifies articulation between financial and non-financial performance. Permits a holistic vision of company strategy. Identifies points where internal control can be improved. Improves administration of risk. Improves efficiency in productive activities. |
| External | Provides information in a world with a growing number of socially responsible investors. Permits the preparation of indicators of sustainability for interest groups. Grants structured non-financial information used in specialised services. Permits greater credibility and valuation of interest groups. Improves brand reputation and diminishes reputation risk. |

Source: Adapted from [55].

Ref. [56] proposes that there are no conclusive results in terms of the value of the IR in the evaluation of management done by stakeholders. Along these lines, [57] finds that the adoption of the IR does not statistically affect the relationship between companies' market value and book value and, at times, does it inversely. These contradictory results could be explained by that proposed by [58] and [59], who demonstrate that the environmental risks and liabilities that were neither recognised nor reported by companies are now considered and evaluated by interest groups and, in consequence, the value of the net assets decreases when it was revealed through the IR.

It is relevant to highlight the virtuous effects of the IR, as when the quantity of information is increased this allows for a better evaluation, in such a way as to improve the

decision-making process [60]. However, the elaboration and revelation of more company information also brings costs with it, classified into direct and indirect costs. We will understand direct costs as those related to the process of collection, classification, dissemination and control of the information. These are easily identifiable and measurable. On the other hand, the indirect costs are the difficult-to-measure hidden ones, related to the revelation of strategic information that can be used against the company by current and/or potential competitors [60]. For more detail of the disadvantages and/or risks of revealing information, see the summary in Table 4.

**Table 4.** Disadvantages of revealing information.

| Dimension | Disadvantage |
| --- | --- |
| Competitive | Revealing profitability for segments of the operation and/or development of new products or technological innovations could encourage current and/or potential competition to develop similar strategies in the short term [33,61]. |
| Legal costs | Even though there is evidence that concludes that a greater level of information revelation diminishes litigation costs [62,63], Ref. [64] affirms that legal costs are one that most concerns managers, due to the possible effects of future information being used by the competition or interest groups. In this vein, Ref. [65] proposes that legal costs are related to the proportion of greater information and when this attempts to give a false impression of the company or unjustified optimism. |
| Political costs | Giving false information can be utilised by other users, such as governments, unions, consumers, clients or providers that could cause pressures on the company, the establishment of greater taxes and new legal and/or environmental demands [66]. |

Source: Adapted from [60].

## 3. Methodology and Sample

This study has a qualitative approach with a phenomenological focus, as it attempts to comprehend the object of the study from the point of view of those who experience it [67]. By the range of its objectives, it is descriptive-type research with a transversal temporal sequence [68], and the data is primary or from a primary source, as it was collected by the authors.

In terms of the sample and the sampling strategy, this work was done in companies listed on the stock exchange in Santiago, Chile. In this respect, it is important to consider that the stock market in Santiago contains several share indices, the most important of which is the SP IPSA, which includes the 29 companies with the greatest share market presence. All the companies, moreover, are regulated by the Financial Market Commission, CMF, a body that obligates them to report aspects of sustainability to the market through an annual statement. The CMF norm does not oblige them to adhere to any specific format to put the statement together and thus, Chilean companies are free to adopt the form they esteem convenient. In this context, of the 29 companies that are included in the SP IPSA, only 11 actively use the format of the IR promoted by the IIRC, which corresponds to the population or universe of this study. The rest of the companies mainly use the format proposed by the Global Report Initiative, GRI or their formats.

The strategy of our sampling was non-probabilistic or intentioned, where the participants agreed voluntarily to be interviewed [69]. Following this, the 11 companies that emit the IR were contacted, but only 4 of them responded affirmatively and agreed to participate in this research.

For this reason, the sample was composed of Banco Itaú-CorpBanca, CMPC, Colbún and Parque Arauco. Additionally, and to enrich the analysis, the Masisa company was invited to participate, because, while it is listed on the Santiago stock exchange and is regulated by the CMF, it is not included in the SP IPSA. It does, however, have the merit of being the first Chilean company to adopt the IR and one of the first in the world to

participate in this reporting initiative. Thus, the unit of analysis of this study are the companies that have adopted the IR in Chile; the unit of information was the executives in charge of the IR in each of the companies that make up the sample [68]. In the following (Table 5), a summary of the companies included in this article is presented:

**Table 5.** Companies included in the study.

| Company | DJSI Chile | IPSA | Integrated Reporting |
| --- | --- | --- | --- |
| Banco Itaú Corpbanca | Yes | Yes | Since 2018 |
| CMPC | Yes | Yes | Since 2016 |
| Colbún | Yes | Yes | Since 2015 |
| Parque Arauco | Yes | Yes | Since 2017 |
| Masisa | No | No | Since 2011 |

Banco Itaú-CorpBanca: The fourth-largest private bank in Chile with a banking platform for future expansion in Latin America, specifically in Chile, Colombia and Peru. Itaú Corpbanca is a commercial bank with headquarters in Chile, and additional operations in Colombia and Panama.

CMPC: A world leader in the production and commercialisation of forestry products, cellulose, paper, tissue products and packaging products. It possesses industrial operations in Chile, Brazil, Argentina, Peru, Uruguay, Mexico, Colombia and Ecuador.

Colbún: Dedicated to the generation and commercialisation of electric energy in Chile and Peru. In Chile, it owns 24 power stations and is the second-largest generator in the country. In addition, Colbún has an installed capacity of 565 MW in the National Interconnected Electric System in Peru.

Parque Arauco: This company develops and administers multi-format real estate assets, mainly of commercial use, oriented to different socio-economic sectors in Chile, Peru and Colombia.

Masisa: A South American company based in Chile, focused on the development of solutions for furniture and interior spaces, sawn lumber, and moulding in the Americas. It also counts on productive capacities to complement its value offering. It possesses a commercial presence in the whole continent and industrial operations in Chile, Mexico and Venezuela.

The instrument of data collection used was that of semi-structured interviews or interviews based on a script [70], which was applied to the executives in charge of the IR in their respective companies. Due to the socio-sanitary contingency in the country because of COVID-19, the interviews were developed through a video-conferencing platform at the end of the year 2020 and the beginning of 2021. These interviews had a duration of approximately one hour, were carried out in Spanish and were recorded to be later transcribed in their entirety. Finally, these transcripts were sent to the interviewees to check that there were no errors in the content and show their conformity with the instrument. The protocol, script or guide for the semi-structured interviews considered three large topics: (a) motivations to emit the IR, (b) perceived benefits from the emission of the IR, and (c) perceived difficulties from the emission of the IR.

The textual material collected was interpreted using qualitative "thematic"-type content analysis, a technique founded on decomposition and classification of the discourse [71]. The management, classification and coding of the text were conducted using the Atlas.ti software in the 9.1.3.0 version.

A limitation to the methodology used is that the qualitative methods are not oriented to generalize their results, but rather to deepen the aspects studied, through the description of the phenomena through their particular features, as perceived in their context, therefore, they do not pretend to measure but to qualify these findings. However, this does not limit the fact that the investigative process can be replicated in other contexts [69]

All the participants received a page with information about the project and the informed consent form by email before the beginning of each interview, in which the ob-

jectives and the methodology were explained, contact information for the researchers was given and the ethical guarantees of the participants were detailed. These guarantees indicate that the participation was voluntary; that the participants could withdraw at any time; that there were no correct or incorrect answers; that the identifiable personal information would only be known by the research team to assure anonymity; that they could end their participation at any time; and that their participation would not imply any risk. Furthermore, it was declared that the participants have no conflict of interest.

## 4. Results

Why emit an integrated reporting?

To understand the reasons that explain the early adoption of the IR in the companies analysed in a voluntary context, it is necessary to make a first classification, alluding to internal and external reasons.

In terms of the external motivations, the need lies in the growing importance of interest groups in the management of the organisations and for which reason improving this relationship becomes vital to be successful in the market. In the opinion of the interviewees, "the Integrated Reporting is a document that speaks about the company and is available for the market, interest groups, and investors". Additionally, the IR constitutes a way for formal corporate communication, and thus, this document is where the effort is made to publicly place all the performance of the company, so that any person, any investor and any regulator that needs to find information can consult it and feel at ease with finding detailed performance information. On the other hand, the IR is perceived as a tool that enhances the corporate image of the organisations.

The interviewees manifested that that emitting the IR "gives a certain status in some circles", above all, those made up of companies committed to sustainability. To complement, another interviewee manifested that, in their industry, "it becomes indispensable" to emit the report. In the same vein, one company manifested that the IR is a worldwide trend, explaining with this the adoption of the format.

On the other hand, there are reasons of an internal nature that explain the adoption of the IR. Along these lines, the emission of the document obliges the company to carry out an in-depth analysis of how the company generates value and knowledge, which is transformed into a powerful management tool. In this way, the IR becomes a traceable, measurable and auditable document. In general, the decision to emit the report goes through a strategic decision. For example, it is indicated that it is the board that signs the declaration of responsibility in each report and thus, there exists an evident commitment with what the report contains and its importance in the strategy.

Additionally, the adoption of the IR is associated with the implicit advantages in the format provided by the IIRC. Proof of this, is that one of the interviewees indicated that "the format of the IIRC is the neatest way to present the information". It is indicated, furthermore, that the format of capitals that the IR uses "allows for showing the business most holistically from all its branches and how it generates value", i.e., it is capable of demonstrating how the use of capitals generates value for the company and, in this context, the format proposed by the IIRC allows it to be presented in a more graphic and comprehensible way.

As a complement, to put together a statement under the IIRC standard obliges the company to think about how to integrate sustainability into management, not just in one area, but in all areas of the company, which translates into an important advantage of the format.

In the same vein, the IR is perceived as a tool that allows for improving performance in aspects of sustainability. This fact, by itself, is established as a category of motivations to adopt the format of the report. One of the interviewees indicated that the idea that "if the business is not sustainable, it is not business" has been gaining ground, which implies a profound change given that, before, anything to do with sustainability was seen

as something like an annexe, with financial-type objectives taking precedence. Adopting the IR is part of the fact that this vision has been changing over time.

The IR places decisions and economic, social and environmental developments, at the same level. That is to say, any decision, be it business, production, market relations, products, or others, obligates an analysis of the obtaining of results not only from an economic point of view but also from a social and environmental perspective. In this vein, one of the companies indicated that "We are fully convinced that, in the long term, it is not possible to generate good economic results if there is not performed with excellence in social and environmental aspects, for which reason sustainability has been integrating into all areas of the business". This is reduced to the fact of having the awareness that the business in which the company operates has an environmental and social impact that is necessary to integrate with the financial aspects. Thus, the IR fulfils this integration. A summary of the motivations behind adopting the IR can be observed in Table 6.

**Table 6.** Motivations behind adopting integrated reporting.

| Categories | Objective |
|---|---|
| Stakeholders | The emission of the IR is oriented towards communication with the distinct stakeholders. It is emitted to have a better relationship with the market. |
| Management | The emission of the IR is oriented towards the interior of the organisation, centred on the benefits the format implies and obeying strategic decisions. |
| Sustainability | The emission of the IR is in accordance with the importance the organisation grants to topics of sustainability. |
| Format | The decision to emit the IR is given by the benefits the format offers. |
| Corporate image | The emission of the IR is oriented towards generating a differentiated corporate image within the market. |
| Worldwide trend | The decision to emit the IR is due to the fact that the format is a worldwide trend. |

What benefits do you perceive with the emission of integrated reporting?

Undoubtedly, the adoption of the IR over other models of corporate reporting generated benefits that have already been described in the literature. In the case of the companies analysed, it has been possible to identify eight categories of benefits perceived with the adoption of the IR. These are summarised in Table 7.

**Table 7.** Benefits of integrated reporting.

| Category | Benefit |
|---|---|
| Costs and efficiency | The adoption of the IR generates savings in costs and efficiency of processes. |
| Decision making | The IR is a tool used in making and managing decisions. |
| Transparency | The IR improves transparency with different interest groups. |
| Sustainability | The IR enhances the inclusion of sustainability in the business. |
| Simplicity | The IR is a simple, easy-to-understand model. |
| Global vision | The IR permits a global vision of the business. |
| Specialised bodies | The IR facilitates the work of analysts, international indices and ranking. |
| Benchmarking | The IR allows comparison with the industry, be it national or international. |
| Legal requirements | The IR makes it possible to meet international legal requirements. |

Concerning the benefits in costs and efficiency, the majority of the companies point out a perception of a reduction in their costs, as "this allowed us to gain in internal efficiency and cost reduction because only one document is generated". In addition, evidence is demonstrated of a certain saving in costs and efficiency by working in a team to not duplicate the gathering of information. Finally, consolidating a financial statement and a sustainability report in an IR generates efficiencies, considering the number of working hours and administrative costs, such as the designs and editing, among others, that are involved in the process.

One point in which all interviewees coincide is in the benefits in the decision-making process that the IR provides in the organisations. Along these lines, one of the companies indicated "the IIRC is the most utilised standard, which allows a better alignment and to generate synergies". In the same way, it is indicated that the IR can consolidate information and thus, accelerate providing information and making decisions. In the same argument, the report is understood as a source of very valuable information as the exercise itself of gathering information allows for the visualisation of figures, data, and management that, during the year, there was no time to evaluate or analyse. Thus, it is a process that gathers information, it is transformed into a retrospective reflection exercise. Based on this, the IR is understood as a management document.

In this way, the IR is indicated as "a working tool of management, that serves the different business units and affiliates in the daily work". Additionally, the report is permanent material to be consulted for the company's main executives and the main leaders of the organisation and helps identify gaps and generates challenges around improving performance. The interviewees indicated that they have the perception that the document becomes a very good repository of content for the companies and finally settles many internal discussions.

In other words, the IR is a document that the whole organisation relies on for its management. One company indicated "there is no area or affiliate of this company that does not have participated in the elaboration of the report, be it directly or indirectly, so all areas can find useful information in the report". In effect, the report helps the company to question its process of value creation and how it is using the different capitals. In all, the IR is perceived as very beneficial for internal processes.

Along the same lines, the IR offers a global vision of the business, and for this reason, is perceived as an important benefit by the interviewees. One of them indicated that "the format of capitals allows us to show the business more holistically from all its branches and how value is generated, with the IIRC format as the most graphic presentation of this effect". At the same time, it is feasible to present just one story of the company, coherent with the integration of sustainability in the business and the business model and responding to the use of the different capitals.

In the area of sustainability, there are varied benefits indicated by the companies. Among them is that the report is indicated as allowing "greater coherence in the story of the integration of the business with sustainability", sending a message to the market that says that the results are not just financial, but also social and environmental. In this line, the advantage of the format promoted by the IIRC is to help the companies integrate sustainability into management, which otherwise could seem academic or theoretical.

On the other hand, benefits are observed as associated with interest groups and the positioning of the companies at a national and international level. Referring to the first concept, the IR becomes a tool to consult and, in this way, interest groups better value the work the companies do. In effect, the report is perceived as a good instrument in which the shareholders and investors seek official information of the companies that finally makes it a set of elements that generate value about the companies and stakeholders. Moreover, and in terms of the positioning of the companies thanks to the IR, in the words of one of the interviewees, "it allows us to be evaluated by a series of international indices that are valued by investors", basically for the fact that the format unites certain international principles and certain recognised forms of reporting.

Besides, it is indicated that there are rankings that review the information published in the IR, as the format facilitates giving information. It is also indicated that the IIRC format facilitates the work of international analysts and the incorporation of the company into indices, such as the Dow Jones Sustainability, because all the information is in one document available to the public.

Finally, there are certain benefits less frequently associated with the emission of the IR, such as the ease of comparison with the industry at a national and international level and the fact that the IIRC format is useful to meet the legal requirements imposed by the Chilean regulator.

What difficulties has the process of adoption of the integrated reporting met with?

Like all new business processes, this one has not been free of difficulties. Firstly, the interviewees indicated problems related to the IIRC standard each time the International Framework, which regulates the presentation of IR, is quite general. In this context, general guidelines are an insufficient standard to generate the report, in the judgement of the companies, and make it necessary to complement with other standards, such as the GRI, which is both more complete and more specific. Additionally, the interviewees indicate "if it were not for the GRI standard and it was necessary to organize the information about capitals, an ocean of information would be left that we would not know how to categorize".

The little knowledge the local market has about the IR format was also indicated as problematic, concerning interest groups and the lack of specialists. Concerning the former, one interviewee indicated, "the investors, the market, the society do not value or understand the document and end up separately making their forms". On the other hand, there exists the perception that the final consumer does not intend to reward the most transparent companies, fundamentally through not knowing and being more susceptible to publicity than concrete information. The companies see this as "an incongruence and deficit of the IIRC and the regulatory bodies, as it does not work to establish more requirements in the statements if, finally, the actors in the market end up requesting filling out their questionnaires".

In terms of the relationship with interest groups, it is perceived as an opportunity to improve the feedback that should be given to the stakeholders about how they were considered in the IR. To close the cycle with them is perceived as a latent difficulty. It is also indicated that there is a lot of disparity of professionals in charge of issues of sustainability in the companies; depending on where sustainability is located, there are focal points in different topics: marketing, people, environment, or operations, but not necessarily as holistic visions. This makes the application of the IIRC model difficult.

As difficulties of the organisations themselves, the process of obtaining information, the internal coordination, the definition of materiality and the costs associated with the report were outlined. In terms of the obtaining of information, one of the interviewees indicated, "we have often spoken internally about how to manage to set up a management system that groups the information easily". Thus, the greater difficulty is in the lack of integration between the material issues and the lack of a management system that can extract the data from the different areas more easily. On the other hand, difficulties are perceived in internal coordination, with the great challenge of achieving an alignment of the whole organisation to report the issues that impact the business. In the same vein, it is indicated that, normally, the teams are receiving many requests for information of different types, which requires additional time for the areas to relate to each other, reading not only their information but also that of other areas. Currently, the everyday work does not facilitate spaces to generate this integration, and this represents a challenge for the companies that use the IR.

Another relevant issue, from the internal point of view, is given by the definition of materiality, or the decision about the issues that will finally be included in the report. Given the amount of information that tends to be collected for the creation of the IR, companies face a complex selection process. In the words of one company, "prioritization of what will finally be considered is quite complex", as it generates legitimate expectations in all of the

interest groups considered. Finally, one interviewee indicated that he considers that the costs associated with the creation of the IR, which is by nature a complex report, imply an economic strain for the organisation. The difficulties mentioned have been synthesised in Table 8.

**Table 8.** Difficulties in the adoption of integrated reporting.

| Category | Difficulty |
| --- | --- |
| Standard | The IIRC standard is tied to the interpretations of each company. |
| Implementation cost | The cost of emitting the IR is higher than other similar reports. |
| Recognition by the market | The IR does not count on important recognition by the market. |
| Obtaining information | The gathering of information for creating the IR is difficult and complex. |
| Materiality | The definition of the material issues to include in the report generates problems. |
| Internal coordination | Emitting an IR implies a considerable internal coordination in the whole organisation. |
| Relationships with interest groups | For the emission of the IR, it is necessary to relate more deeply with different interest groups. |
| Lack of specialists | There is no critical team of specialists in IR. |

## 5. Discussion

As indicated, the objective of this work is to analyse the early adoption of the IR in an emerging economy, in this case, the Chilean economy, with an emphasis on the reasons for this adoption, the perceived benefits and difficulties experienced. In this section, the results obtained will be compared with those of the existing literature.

In the first place, within the motivations for adopting the format of the IIRC, an improvement in decision making was indicated and this is confirmed given that this concept is also repeated in the benefits of its adoption. This situation is consistent with [60]. In the same vein, it is necessary to consider that the adoption of the IR must not be limited to a change in format, but also in purpose. It is to positively affect the internal processes and organisational structure of the companies that drive value creation [9,10]. In the same way, [55] indicates, as benefits, a better comprehension by the organisation of its performance indicators, and the identification of points where internal control could be better, all of which are concepts associated with an improvement in the internal management of the organisation. Finally, [72] indicated that adhering to the format of the IIRC generates management processes that are more integrated and clearer in the issues of the business and performance, and [45] pointed out that there exists evidence that shows that the decision-making process improves through the use of the IR.

Another point, highlighted as motivation, benefit, and also the difficulty, is the relationship with the related parts. Along these lines, it is important to keep in mind that the demands in the preparation and presentation of the IR bring with them, implicitly, a change in attitude and relationship of the organisation with its stakeholders [10,37,42,46]. This situation has been described by [55], who indicated that the adoption of the IR allows the preparation of useful indicators for the different interest groups. Along the same lines, [72] established an improvement in corporate reputation and relationships with the interesting parts. To complement this, [73] provided evidence of improvements in the relationship with the interesting parts.

Equally, the holistic vision that the IR provides of the business is indicated as one of the benefits. In this perspective, [74] indicated this as a key element in the adoption of the IIRC, and that is also coherent with [73], who pointed out that the IR is a global document

that combines different streams of corporate information, such as financial, governance and sustainability information.

On the other hand, an important concept is an improvement in the corporate image of the company or how the companies are perceived in the market for the use of the IR. This element was indicated as a motivation and a benefit. This is consistent with that indicated by [55], who defined an improvement in brand reputation and a decrease in reputation risk, which is similar to that indicated by [72] or [75], who identified that the IR was an important part of the credibility and legitimacy of the organisations.

It must not be forgotten that the IR is born in a context of integration of sustainability into business models, and for this reason, the concept of "sustainability" has been key in the findings of this research. The format of the IIRC provides information in a world with a growing number of socially responsible investors [55]. In the same way, [74] pointed out that the adoption of the IR is present in companies that have already observed the importance of sustainability in their business models.

Due to the novelty of the concept implicit in the IR, the companies still have no guarantee of the objectives of the adoption and the efficacy of the report [74]. This situation was also demonstrated in this analysis, given that many concepts are identified in the existing literature, but others are not, for example, the dispersion among professionals, specialists in IR and the costs of implementation itself.

Finally, the format of the IIRC catches the attention. It is perceived as a benefit, but also as a difficulty, due to how generic it is. In this sense, [26] argued that for a group of Colombian companies, the IR is interested in the risks of the organisation on operating in the zones where it develops its activities, as opposed to the GRI format that centres on the impact of the company on its surroundings. It is logical to question whether the IIRC format is an appropriate standard in an emerging market, or if its adoption responds to the existence of isomorphic pressures of mimicry that would explain the adoption of new standards of company information [76], as indicated by institutional theory. This can be supported by the fact that all the interviewees indicated the improvement in corporate image and the possibility of being comparable at a national or international level, be it by the market itself, by analysts or by specific indices of sustainability, among others, as advantages of the report.

## 6. Conclusions

The IR has become a strong global trend with strong global growth since the year 2011 [11] and thus, this phenomenon is already present in a large part of the world, including emerging markets, such as that in Chile.

While the majority of the results are from the literature, there are diverse nuances given the context of an emerging economy. There is great importance towards the outside world, with the relationship with interesting parts, better perception by the market and inclusion in an international context as the most reiterated concepts. It is also worth noting that all the companies analysed declared sustainability as a strategic focal point, which is an unusual situation in developing markets, such as in Chile. Underlying this declaration is a paradigm shift in Latin American companies, as has already been described by [26].

The contradiction with the format of the IR is also worth noting. On one hand, the format of the IIRC is indicated as a powerful reason, given its simplicity and possibility of seeing the organisation as a whole but, at the same time, it is indicated that the format is lax, with possibilities of making ones' own interpretations, which is indicated as a weakness in the standard. The logical question is to determine if the national companies really, fully understand what the IIRC format proposes, or if they adopt only part to be part of a worldwide trend. The analysis suggests that the companies have observed concrete benefits with the IR and, maybe, only a natural maturing process is necessary. Along these same lines, it is necessary to complement the fact that there exists heterogeneity in the professionals who work or consult in the creation of the report.

Despite the conclusions, it is important to consider that this work has different limitations: fundamentally, the low adhesion to the format of the IR in Chile and, thus, the low valuation that market agents still have towards it. In this same vein, the format will likely be boosted with the emission of the General Character Norm 461-2021 from the CMF, which obligates companies that are held accountable to emit information and divulge their financial and non-financial information. Another limitation of this research is that the methodology used does not intend to generalize the results obtained; however, this does not limit the fact the investigative process can be replicated in other contexts, which allows new research opportunities. However, despite this, this work is a contribution to the global discussion about IR.

In light of the above, the main contribution of this article is that it has been carried out in an emerging economy in companies with relevant experience in the adoption of IR, as well as the perception of the reasons, benefits and difficulties in the preparation of the integrated report. This is important because the Chilean market provides information to different stakeholders on the sustainability of its operations in the short and long term. Furthermore, this paper contributes to the literature related to corporate governance concerning the use and value of integrated reporting. This is so, even though there are several similar works concentrated in developed economies that are driven by other incentives and motivations. IR research in emerging markets is a nascent field for academic study. Despite methodological limitations, this work can be replicated in other companies in Chile, as well as in other emerging markets, and promote the use of integrated reporting

Finally, this work opens lines of future research, such as in the opportunities for improvement that the preparers see in the current format of the IIRC, in the analysis of internal changes these organisations have had to carry out to face the challenge of the IR and concerning interest groups and processes of materiality.

**Author Contributions:** Formal analysis, J.N.-O.; Investigation, H.M.-F.; Methodology, J.A.G.M.; Writing and editing, J.L.G. All authors have read and agreed to the published version of the manuscript.

**Funding:** This research received no external funding.

**Institutional Review Board Statement:** Not applicable.

**Informed Consent Statement:** Informed consent was obtained from all subjects involved in the study.

**Data Availability Statement:** The data presented in this study are available on request from the corresponding author. The data are not publicly available because they contain private information.

**Acknowledgments:** Francisco Torrealba (CMPC), Ana Luisa Vergara (Colbún), Catalina Rodriguez (Banco Itaú), Ricardo Vargas (Masisa) and Ximena Bedolla (Parque Arauco).

**Conflicts of Interest:** The authors declare no conflict of interest.

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
