# Peer review of "Why Integrated Reporting? Insights from Early Adoption in an Emerging Economy"

_sustainability, doi:10.3390/su14031695_

Round 1
Reviewer 1 Report
No further comments.
Author Response
Dear reviewer:
We would like to thank you for your review, which has substantially improved our work.
The authors

Reviewer 2 Report
- line 55 there is 14-15-16-17, I think it should be 14-17
- lines 50, 60 - dots are missing
- there is [27-28-29-11], I prefer [11,27-29] etc.
- Is this correct: Table N°7 ?
- Please be aware that you use case study analysis so be careful with the generalization of your research findings. I suggest adding the limitations section.
- Please explain why you have chosen Chile. Is it a good representation of emerging economies?
- I can not see research questions or hypotheses?
Author Response
Dear reviewer:
We would like to thank you for your comments, which have substantially improved our work.
Please find attached responses to your comments.
The authors

Reviewer 3 Report
The aim of the paper is to analyze how the adoption of the Integrated Reporting influence companies from an emerging economy, specifically, Chilean economy. For that, a series of semi-structured interviews to companies from the Chilean stock exchange were made. The results obtained are interesting, however, there are some changes that need to be done in order to improve the paper.
The paper should conform to the requirements of the journal. For example, the abstract exceeds the 200-word limit established in the journal's guidelines.
The introduction should develop in more depth the reasons why the analysis is made, this is, the gap in the literature that the paper tries to cover. In addition, authors should explain what the contributions of the paper in more detail are.
In the conclusions, the limitations of the work should be developed in more detail, since only the limitations related to the IR in Chile are mentioned. The limitations of the method used should also be discussed, as well as possible future research using other methods of analysis. The contributions of the paper should also be developed in more depth, as well as the practical and theoretical implications of the paper.
The format of the references should be revised and standardized. For example, there are references that contain DOI and others do not.
Author Response

(The authors gave the same response as above.)

Round 2
Reviewer 3 Report
I am pleased to read this new version of the paper. The requested changes have been made increasing the quality of the paper. So, the paper can be considered for publication.
Author Response
Dear Reviewer:
Please find attached our response to your final comments.
The Authors
